# Three Rounds of Read Correction Significantly Improve Eukaryotic Protein Detection in ONT Reads

**DOI:** 10.3390/microorganisms12020247

**Published:** 2024-01-24

**Authors:** Hussain A. Safar, Fatemah Alatar, Abu Salim Mustafa

**Affiliations:** 1OMICS Research Unit, Health Science Centre, Kuwait University, Kuwait City 13110, Kuwait; hussain.safar@ku.edu.kw; 2Serology and Molecular Microbiology Reference Laboratory, Mubarak Al-Kabeer Hospital, Ministry of Health, Kuwait City 13110, Kuwait; falatar@outlook.com; 3Department of Microbiology, Faculty of Medicine, Kuwait University, Kuwait City 13110, Kuwait

**Keywords:** eukaryotes, ONT, read correction, gene detection, protein annotation

## Abstract

Background: Eukaryotes’ whole-genome sequencing is crucial for species identification, gene detection, and protein annotation. Oxford Nanopore Technology (ONT) is an affordable and rapid platform for sequencing eukaryotes; however, the relatively higher error rates require computational and bioinformatic efforts to produce more accurate genome assemblies. Here, we evaluated the effect of read correction tools on eukaryote genome completeness, gene detection and protein annotation. Methods: Reads generated by ONT of four eukaryotes, *C. albicans*, *C. gattii*, *S. cerevisiae*, and *P. falciparum*, were assembled using minimap2 and underwent three rounds of read correction using flye, medaka and racon. The generates consensus FASTA files were compared for total length (bp), genome completeness, gene detection, and protein-annotation by QUAST, BUSCO, BRAKER1 and InterProScan, respectively. Results: Genome completeness was dependent on the assembly method rather than on the read correction tool; however, medaka performed better than flye and racon. Racon significantly performed better than flye and medaka in gene detection, while both racon and medaka significantly performed better than flye in protein-annotation. Conclusion: We show that three rounds of read correction significantly affect gene detection and protein annotation, which are dependent on assembly quality in preference to assembly completeness.

## 1. Introduction

Oxford Nanopore Technology (ONT), a third-generation sequencing technology, serves as a platform to sequence small to large and multiplex genomes and is currently widely used globally, especially in low- and mid-income countries, due to its simplicity, feasibility, and sustainability in both medical research and clinical settings [1,2]. The main advantage of ONT is the generation of real-time analysis using the user-friendly interface, EPI2ME Agent, with no bioinformatic expertise required, allowing rapid and fast detection of microbe identification and antimicrobial resistant genes (AMR) [3,4]. The agile and simple library preparation for ONT sequencing without the biased PCR amplification step is another significant advantage [5]. Furthermore, ONT overcomes the problems observed in next-generation sequencing (NGS) in sequencing genomic repeats and the production of incompletely assembled genomes [6]. ONT sequencing generates ‘long-enough’ reads to exceed the length of repeated regions and generates near-complete assemblies in which the location of resistant genes can be detected—i.e., chromosomal vs. plasmid [7,8]. 

Despite the advantages of ONT and the rapid advancement of the technology since its development, the major shortcoming of this technology is the production of relatively high error rates (~10–15%) compared to NGS, when using R9 flow cells [9]. Although increasing the depth of ONT reads can produce contiguous assembled genomes, the errors accumulate as the sequencing depth increases [10]. ONT reads often require read correction with short reads to generate complete and robust genome assemblies. The hybrid genome assemblies produced using both long and short sequencing reads (with sufficient depth of both short and long reads), enhance the accuracy of assembled genomes for downstream analysis [11]. However, having access to both long and short sequencing platforms and the performance of two sequencing experiments on a single sample is impractical—especially in low- and middle-income countries and in clinical settings where prompt diagnoses are important. Therefore, there is a need for alternative low-cost methods to obtain more accurate genome assemblies from ONT reads.

Computational and bioinformatics tools analysing ONT reads are freely available and rapidly expanding. These tools can be counted as a reasonable and low-cost option to reduce error rates post-assembly. These tools use varied algorithms that are designed to identify and resolve sequencing errors to not only produce a complete but also an accurate genome assembly, though the output of the read correction step is reliant on the applied methods and their specific parameters [12]. Several studies are benchmarking freely available read correction tools and their impact on downstream analysis [13,14,15,16]. Among the several available read correction tools, flye, medaka and racon are most commonly used for ONT reads. While flye read assembly and correction tool is based on the generalized Brujin Graph, medaka and racon are tools created to outer-perform graph-based methods generating genomic consensus in much faster time [12,13,16]. The process of benchmarking freely available read correction tools holds significant importance within the scientific community as it plays a pivotal role in advancing the research domain allowing improved analytical precision and resolving critical issues.

Most benchmarking studies focus on prokaryote genome assemblies rather than eukaryote. Whilst ONT has become an important platform for eukaryotic DNA sequencing, allowing an in-depth analysis of complex eukaryotic DNA sequences for virulence factors and gene annotation, there is a need to benchmark the impact of read correction tools on eukaryotic genomes and their downstream analysis.

In this study, we retrieved ONT sequencing reads from the Sequencing Read Archive (SRA)–NCBI of four pathogenic eukaryotes: *Candida albicans*, *Cryptococcus gattii*, *Saccharomyces cerevisiae*, and *Plasmodium falciparum*, and evaluated the impact of applying three read correction tools: flye, medaka, and racon, on genome length, fragmentation and completeness, and accurate gene structure, and analysed and classified eukaryotic functional proteins. The selection of these organisms was primarily motivated by the availability of high-quality sequencing data in the SRA–NCBI database through ONT methods. This choice was further supported by their significance as model organisms, as exemplified by *S. cerevisiae*, and their significance as pathogens.

## 2. Materials and Methods

The sequencing reads (FASTQ) of four eukaryotic species, (n = 6 each), were retrieved from the SRA–NCBI (Appendix A). The sequencing reads were all generated using an ONT ligation sequencing kit (LSK-109) with R9 flow cells. The FASTQ reads were then filtered based on quality (Q score > 10) using NanoFilt (version 2.6.0) [17]. The adapters and read barcodes were then trimmed by Porechop (version 0.2.1) (https://github.com/rrwick/Porechop, accessed on 1 September 2023).

The filtered and trimmed FASTQ reads were then aligned against a reference genome sequence (Appendix A) using Minimap2 (version 2.17-r941) [18]—using default parameters—in combination with bcftools (version 1.5) (https://samtools.github.io/bcftools/, accessed on 1 September 2023) and bedtools (version 2.30) (https://bedtools.readthedocs.io/en/latest/, accessed on 1 September 2023) to remove missing and/or low-coverage sites/nucleotides. Qualimap (version 2.2.2-dev) [19] was used to detect the mapping percentage in the BAM files generated in the minimap2 procedure. Reads with >85% coverage mapping against the reference genome were further analysed (Appendix A). The consensus FASTA files generated went through three rounds of read correction process with flye (version 2.8.3-b1695) with polish-target parameter, medaka (version 0.11.0) (https://github.com/nanoporetech/medaka, accessed on 1 September 2023) and racon (version 1.4.10) with no-trimming parameter [20,21].

The quality of generated consensus FASTA files from minimap2, flye, medaka, and racon (n = 24 per species, n = 96 in total) were assessed by QUAST (version 5.0.2) using the LG parameter. The total length (bp), total aligned (bp), and GC%, were evaluated [22].

The sum of genome completeness, duplication rate, genome fragmentation, and missing genes were evaluated by Universal Single-Copy Orthologues (BUSCO) (version 5.2.2) [23]. Accurate eukaryotic gene structure annotation of the consensus FASTA files was assessed with BRAKER1 (version 3.0.3) with GeneMark-ET. The generated GFF3 files containing complete coding DNA (CDs), forward CDs, reverse CDs, mRNA, and introns were then visualized with pycirclize (version 0.5.1) (https://github.com/moshi4/pyCirclize, accessed on 1 September 2023) [24,25,26,27]. InterProScan (European Molecular Biology Laboratory’s European Bioinformatics Institute) (version 5.63–95.0) was used to fully analyse and classify eukaryotic functional proteins using ProSiteProfiles analysis [28]. All consensus FASTA files, codes, and commands are available at https://github.com/hussainsafar/eukaryotes_read_correction, accessed on 1 September 2023.

Statistical analysis was performed with Bonferroni’s multiple comparison one-way ANOVA by GraphPad Prism (Boston, MA, USA) (version 8.0.1) to determine significant differences (*p* < 0.05, *p* < 0.001) existing among the consensus FASTA files generated by minimap2 before and after read correction with flye, medaka and racon, in QUAST-based assembly statistics, gene and protein detection/prediction by BRAKER1 and InterProScan.

## 3. Results and Discussion

Eukaryotic whole genome sequencing provides comprehensive insights into their complex genomes. ONT sequencing is a practical long-read sequencing platform that enables rapid and cost-effective identification of strains, and detection of virulence factors and proteins in both research and clinical settings. However, the relatively higher error rates produced by ONT reads require computational and bioinformatics efforts to produce contiguous and accurate eukaryotic genome assemblies. In this study, we examined the effect of three rounds of read corrections using flye, medaka, and racon after assembling ONT reads to a reference genome using minimap2. The evaluation was based on the genome total length (bp) and GC% produced by QUAST, genome completeness detected by BUSCO, gene prediction by BRAKER, and protein annotation by InterProScan. We used default parameters and datasets provided by the bioinformatic tools.

QUAST analysis assessed the quality and accuracy of genome assemblies pre- and post-three rounds of read correction. The total length (bp) was significantly (*p* < 0.05) higher after read alignment with minimap2 against the reference genomes than post-read correction of *C. gattii*, *S. cerevisiae*, and *P. falciparum* (Table 1). Nevertheless, the median total length after read correction was the lowest after correction with flye and significantly (*p* < 0.05) improved with the second and third rounds of correcting with medaka and racon, respectively (Table 1). The improvement of assemblies’ total length is a common feature. Studies have reported improvements up to 57% in genome assemblies; however, in this study, we noticed improvements of 9.36% only [8,16]. The variation in improvement percentage depends upon various factors, such as organism sequencing, DNA library preparation, genome assembly, and read correction tools used. Although the total aligned (bp) was highest after minimap2 assembly, it was not significant (*p* > 0.05) (Table 1) when compared to assemblies after read correction. The total aligned length was the highest after the second round of read correction with medaka and was the lowest after the third read correction with racon. The GC% was significantly higher (*p* < 0.05) (Table 1) after read correcting with flye and decreased after the second and third rounds of read correcting. In line with other studies, we previously noticed similar outcomes; although medaka and racon had significantly lower GC%, both read correction tools performed better in the overall genome assembly, especially when combined [16,29,30,31].

BUSCO provides a quantitative measure of genome completeness to evaluate the quality of genome annotation. Among the four eukaryotic species examined in this study, medaka showed improvement over minimap2 only in *C. albicans* assembled genomes regarding genome completeness (Figure 1a). When comparing the read correction tools, medaka was also more superior than flye and racon in genome completeness in all four species samples (Figure 1). While the usage of medaka for diploid cells has been controversial because of the diploid nature of yeast, we found that the newer version of medaka provided more accurate assemblies. These results are in line with Sigova et al. [32]. In their study, they reported that read correction with medaka is superior to read correction with racon in fungal pathogens. In addition, the percentage of genome completeness significantly decreases (by ~40%) when a reference is added, even after using six read correction tools [32]. Moreover, Zhang et al. showed that medaka performance was superior against other read correction/polishing tools in which medaka improved the continuity and reduced mismatches in *S. cerevisiae*-assembled genomes [33]. In all species, except *P. falciparum*, flye was superior to racon in genome completeness and duplication rates (Figure 1). The rate of the fragmented genome was comparable in all species for all three rounds of read correction (Figure 1).

Genome completeness is majorly affected by sequencing methods and genome assembly tools rather than read correction tools [33]. The higher number of genome completeness observed in uncorrected assemblies in this study was due to minimap2 assembly, which is a reference-based alignment method. Other studies using de-novo genome assembly methods show—with sufficient sequencing depth—the advantages of using read correction tools in BUSCO analysis [33,34].

BRAKER1 is a bioinformatic tool commonly utilized for gene prediction in eukaryotic genomes using GeneMark-ET. Ideally, eukaryotic genome assemblies are combined with RNA-seq data to improve gene prediction accuracy. However, the ability to combine both DNA and RNA-seq data is not often available in real scenarios. Here, we performed BRAKER1 analysis on assembled and corrected genomes to evaluate the total number of CDs, forward CDs, reverse CDs, mRNA, and introns (Figure 2, Figure 3, Figure 4 and Figure 5). The total numbers of CDs, forward CDs, and reverse CDs were significantly higher after the third round of read correction with racon (*p* < 0.05 vs. minimap2, *p* < 0.001 vs. flye, and *p* < 0.05 vs. medaka) (Figure 2, Figure 3, Figure 4, Figure 5 and Figure 6). Surprisingly, the total number of CDs increased after the first round of read correction with flye but decreased after the second round of read correction with medaka (Figure 2, Figure 3, Figure 4 and Figure 5). In the samples of *C. albicans*, *C. gattii*, and *P. falciparum*, the total number of CDs after read correction with racon was higher than flye by 55273, 176705, and 63178, respectively. However, the total number of CDs in the samples of *S. cerevisiae* was lower after read correction with racon. The effect of genome assembly and read correction pipelines on the *S. cerevisiae* genome has been well characterised [33]. The authors concluded that although read correction improved contiguity and coverage, sequencing depth and choice of sequencing method affect *S. cerevisiae* genome annotation [33]. The number of introns showed a parallel significance pattern to the total number of CDs. The total number of introns was significantly higher after read correction with racon (*p* < 0.05 vs. minimap2, *p* < 0.001 vs. flye and medaka) (Figure 2, Figure 3, Figure 4, Figure 5 and Figure 6) in the samples of *C. albicans*, *C. gattii* and *P. falciparum*, but not *S. cerevisiae*. Similarly, Shin et al. [35] found that applying the Nanopolish read correction tools to reads assembled by the Canu-SMARTdenovo method increased the detection of CDs and introns when using MAKER2 as an annotation tool. Interestingly, the number of introns after the first round of read correction with flye was significantly higher (*p* < 0.05) than after genome assembly with minimap2 (Figure 6). On the contrary, the number of mRNA coding genes was the highest after genome assembly with minimap2. Among the three rounds of read correction, the highest number of mRNA coding genes was detected after the second round of read correction with medaka, which was only significant against racon (*p* < 0.05) (Figure 2, Figure 3, Figure 4, Figure 5 and Figure 6). Given the size of mRNA coding gene, which is ~1500 nucleotides in average, detecting mRNA coding genes is very critical [36,37]. Like other coding genes, these genes undergo quality control and trimming steps to remove low-quality and/or adapters present in the sequencing reads. Hence, the trimming process by read correction tools can generate even smaller gene sizes which no longer map to the reference genomes in the databases. Although the number of mRNA coding genes was lower after the third round of read correction with racon, this may result from removing all false-positive genes detected post-genome assembly with minimap2.

Based on BRAKER1 gene prediction accuracy results, we investigated the effect of read correction tools on protein annotation by InterProScan with ProSiteProfiles analyses, describing protein domains, families, and functional sites. The overall hits of protein annotation were improved with each round of read correction in all four species, with racon being the top-performing read correction tool (Figure 7a). Several protein annotations were only detected after applying a read correction to the assembled genomes, such as TGF-beta binding (IPR017878), colipase family (IPR001981), and Cytochrome c class II (IPR002321) in *C. gattii* samples; streptavidin (IPR005468), Cytochrome c, class II (IPR002321), and GATA-type zinc finger (IPR000679) in *S. cerevisiae*; and platelet-derived growth factor (PDGF) (IPR000072), coronaviridae zinc-binding (CV ZBD) (IPR000072), GATA-type zinc finger (IPR000679), and C-terminal cystine knot (IPR006207) in *P. falciparum* samples (Figure 7a). Protein annotation hits of IPR002321 detected by medaka were significantly (*p* < 0.05) higher than minimap2, flye, and racon in *C. albicans*, whereas protein annotation hits of IPR00724 and IPR001002 detected by medaka were significantly (*p* < 0.05) higher than minimap2, and protein annotation hits of IPR002321 detected by medaka and racon were significantly (*p* < 0.05) higher than minimap2 and flye (Figure 7b). In *S. cerevisiae* samples, protein annotation hits of IPR007112 detected by racon were significantly higher than hits detected by minimap2 (Figure 7b). Protein annotation hits of IPR001938 detected by medaka were significantly (*p* < 0.05) higher than hits detected by flye in *P. falciparum* samples (Figure 7b).

To our knowledge, this is the first study to evaluate the effect of read correction tools for long-reads on gene prediction using BRAKER1 and protein annotation using InterProScan. Although BUSCO analysis showed superior genome completeness to uncorrected assemblies, we found that read correction tools offer advantages over uncorrected assemblies in BRAKER1 gene detection and protein annotation using InterProScan with ProProfiles analysis. In this study, we showed that genome accuracy after three rounds of read correction is more vital for gene prediction and protein annotation than genome completeness. We proved that gene prediction accuracy relies on the quality of assembled genomes after read correction rather than the quantity or the number of present genes after genome assembly. In other words, a more accurate genome assembly leads to more reliable gene prediction and protein annotation [38,39]. However, the gene completeness analysis could still be improved. The development of more robust read assembly and read correction tools and pipelines is still an area to explore. Studies have shown that the usage of mix-and-matched freely available read assembly and read correction tools significantly improves not only assembly parameters, but also antimicrobial resistant genes detection, plasmid identification and pan-genome analysis with and without using short sequencing reads for read correction [14,16,40,41,42]. In addition, adjusting the read assembly and/or read correction tools parameters could be beneficial. Schiavone et al. [43] has documented the importance of applying ‘tailored’ bioinformatics analysis. Obtaining complete sequences of chromosome and plasmid of *Salmonella enterica* was possible by modifying corErrorRate and corMincoverage parameters in Canu assembler [43].

In addition, improving the sequencing platform itself can reduce sequencing error rates and increase accuracy, which has been observed since the development of ONT from the production of R6 flow cells until now [44]. ONT has recently introduced the flow cells (R.10.4.1) with a quality score >20. The preliminary outcome of these flow cells is very encouraging [45]. The performance of the R10 flow cells outperforms the R9 flow cells, achieving a genome accuracy of >99% [45,46]. However, to achieve near-complete genomes, short reads may still be required for read correction [47]. The performance of the new R20 flow cells is still being investigated, and their combination with different read assembly and read correction tools is yet to be investigated.

## 4. Conclusions

The rapid development of whole-genome sequencing platforms has revolutionised their usage and application in research and clinical settings. Using both short- and long-sequencing reads to produce hybrid genome assemblies is a very robust method for gene detection and protein annotation. However, access to both short- and long-sequencing platforms is an unrealistic scenario, especially in low- and mid-income countries. ONT serves as a reliable and relatively inexpensive long-reading sequencing platform. However, the major burden of this sequencing platform is the relatively higher error rate. Therefore, improving the sequencing reads generated by ONT by computational and bioinformatics tools is a logical and cost-effective option.

Numerous long-read correction tools are regularly generated aiming to achieve robust genome assemblies. These tools often use different bioinformatic algorithms. Benchmarking the freely available read correction tools is very important and drives the research field to better analysis resolution. This study showed that genome quality is more important than genome completeness. Although genome completeness was significantly higher in pre-read correction steps, significant improvement in gene prediction and protein annotation in eukaryotic genomes was noticeable after the second and third rounds of read correction. However, the assembled genomes can still be improved for better outcomes. Therefore, the investigation of several read correction tool combinations is required along with the improvement of ONT-sequencing technology.

## Figures and Tables

**Figure 1 microorganisms-12-00247-f001:**
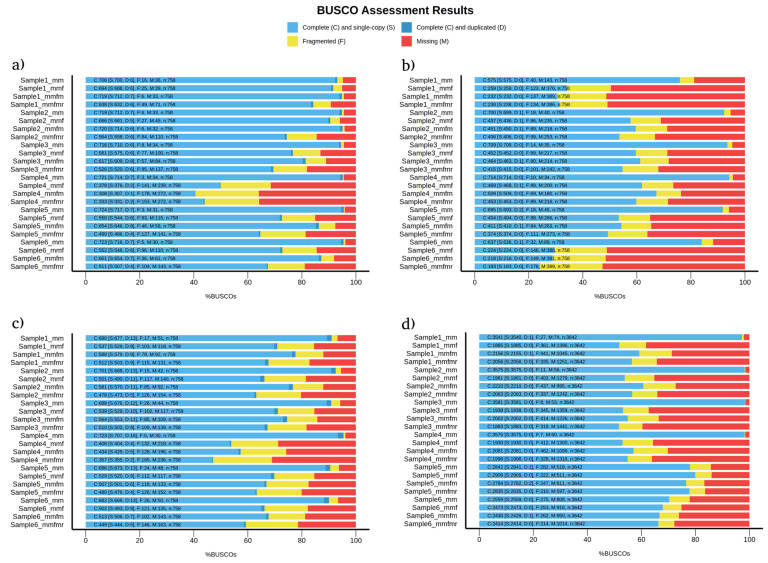
BUSCO analysis detecting genome completeness, genome duplication, fragmented genes, and missing genes in (**a**) *C. albicans*, (**b**) *C. gattii*, (**c**) *S. cerevisiae*, and (**d**) *P. falciparum* samples. mm = uncorrected minimap2, mmf = minimap2 corrected with flye, mmfm = minimap2 corrected with flye + medaka, and mmfmr = minimap2 corrected with flye + medaka + racon.

**Figure 2 microorganisms-12-00247-f002:**
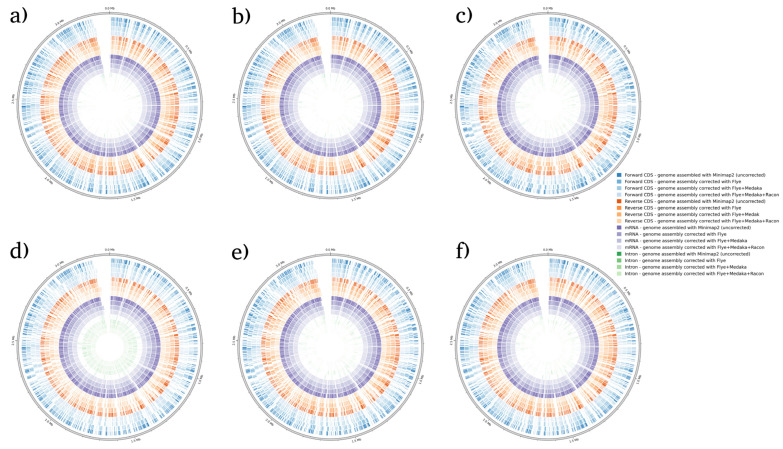
BRAKER1 analysis detecting forward CDs, reverse CDs, mRNA, and intron in *C. albicans* species. (**a**) Sample 1, (**b**) sample 2, (**c**) sample 3, (**d**) sample 4, (**e**) sample 5, and (**f**) sample 6. Bonferroni’s multiple comparison one-way ANOVA statistical analysis was performed to determine significant differences (*p* < 0.05, *p* < 0.001) existing among the different groups.

**Figure 3 microorganisms-12-00247-f003:**
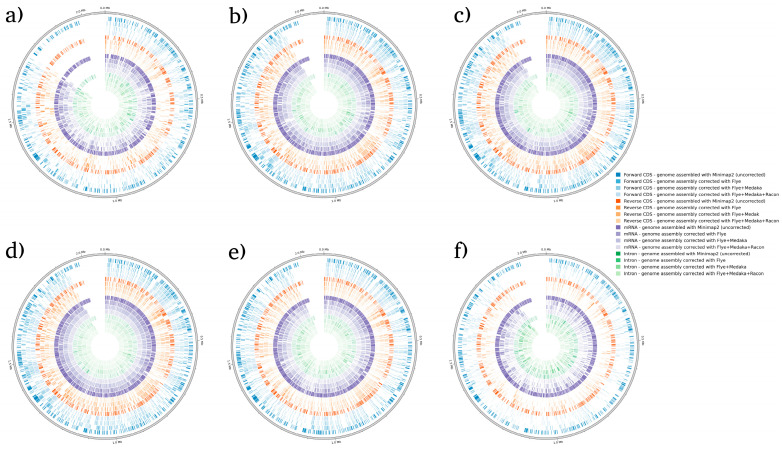
BRAKER1 analysis detecting forward CDs, reverse CDs, mRNA, and intron in *C. gattii* species. (**a**) Sample 1, (**b**) sample 2, (**c**) sample 3, (**d**) sample 4, (**e**) sample 5, and (**f**) sample 6. Bonferroni’s multiple comparison one-way ANOVA statistical analysis was performed to determine significant differences (*p* < 0.05, *p* < 0.001) existing among the different groups.

**Figure 4 microorganisms-12-00247-f004:**
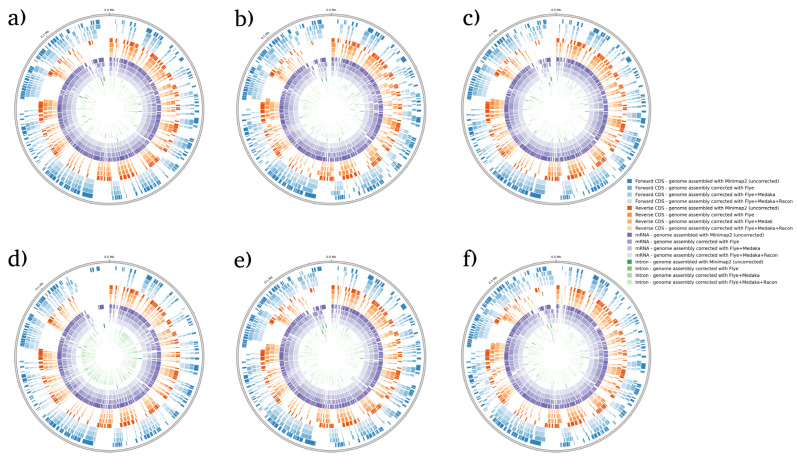
BRAKER1 analysis detecting forward CDs, reverse CDs, mRNA, and intron in *S. cerevisiae* species. (**a**) Sample 1, (**b**) sample 2, (**c**) sample 3, (**d**) sample 4, (**e**) sample 5, and (**f**) sample 6. Bonferroni’s multiple comparison one-way ANOVA statistical analysis was performed to determine significant differences (*p* < 0.05, *p* < 0.001) existing among the different groups.

**Figure 5 microorganisms-12-00247-f005:**
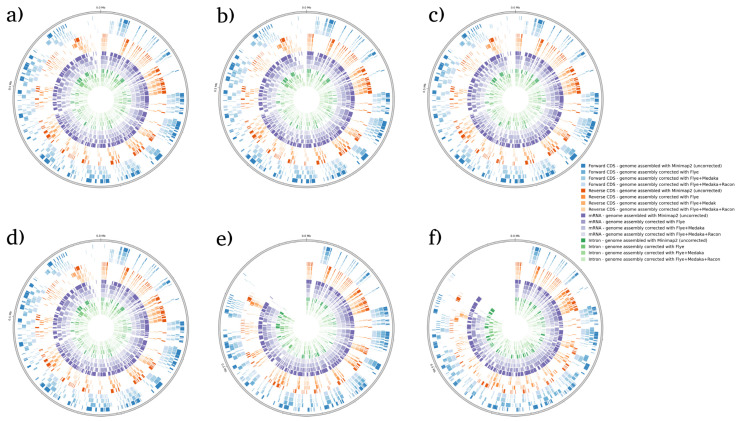
BRAKER1 analysis detecting forward CDs, reverse CDs, mRNA, and intron in *P. falciparum* species. (**a**) Sample 1, (**b**) sample 2, (**c**) sample 3, (**d**) sample 4, (**e**) sample 5, and (**f**) sample 6. Bonferroni’s multiple comparison one-way ANOVA statistical analysis was performed to determine significant differences (*p* < 0.05, *p* < 0.001) existing among the different groups.

**Figure 6 microorganisms-12-00247-f006:**
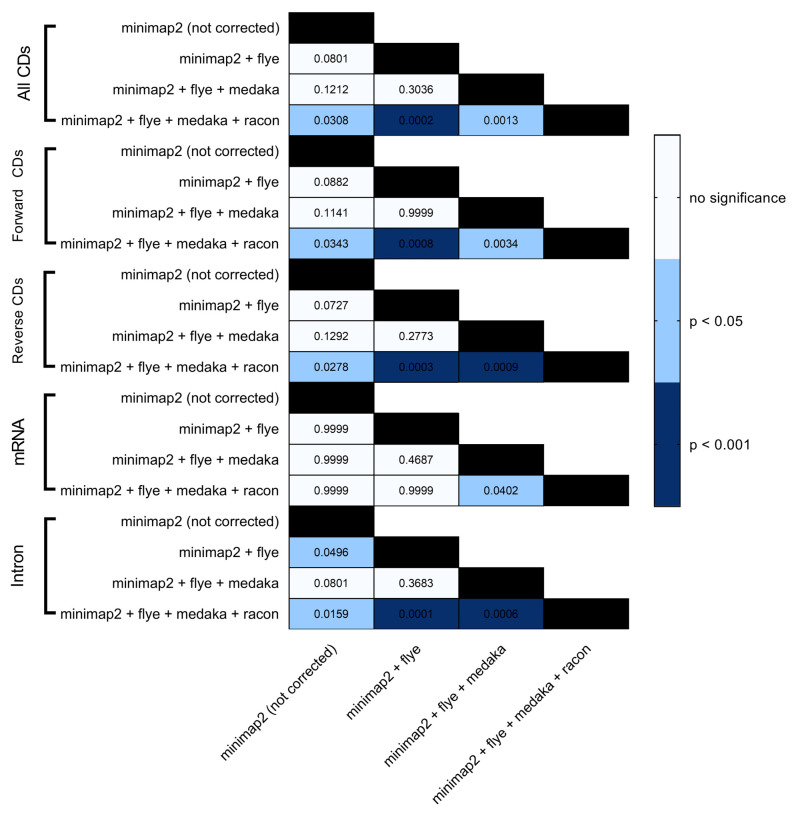
Heatmap statistical analysis for BRAKER1 results. Bonferroni’s multiple comparison one-way ANOVA was performed to determine significant differences (*p* < 0.05, *p* < 0.001) among minimap2 before and after read correction with flye, medaka and racon.

**Figure 7 microorganisms-12-00247-f007:**
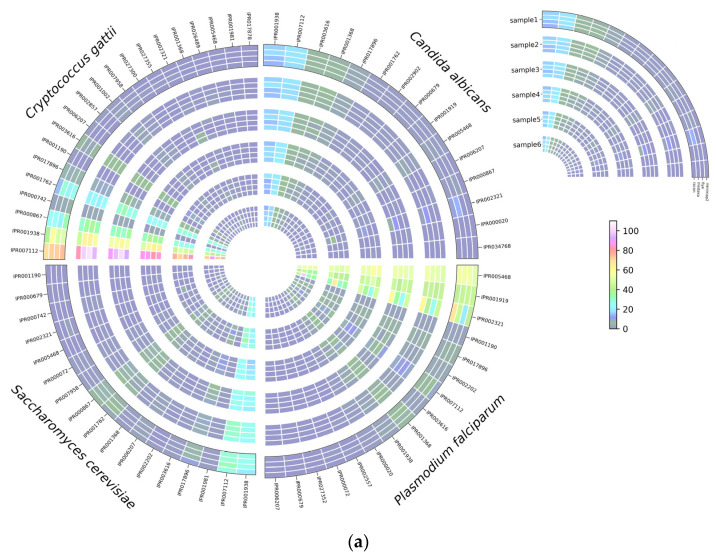
InterProScan analysis using ProProfile analysis for protein annotation in *C. albicans*, *C. gattii*, *S. cerevisiae*, and *P. falciparum*, (**a**) number of hits detected, and (**b**) the significant differences among read correction methods. Bonferroni’s multiple comparison one-way ANOVA statistical analysis was performed to determine significant differences (*p* < 0.05, *p* < 0.001) existing among the different groups.

**Table 1 microorganisms-12-00247-t001:** Total length (bp), total aligned (bp), and GC% of ONT-sequencing reads aligned with minimap2 before and after applying as read correction tools.

Correction Tool		Minimap2 (Not Corrected)			Flye			Flye + Medaka			Flye + Medaka + Racon		
		Total Length (bp)	Total Aligned (bp)	GC%	Total Length (bp)	Total Aligned (bp)	GC%	Total Length (bp)	Total Aligned (bp)	GC%	Total Length (bp)	Total Aligned (bp)	GC%
*C. albicans*	Sample 1	14268731	14255757	33.45	14272767	14231426	33.49	14317735	14250916	33.43	14319429	14255001	33.43
	Sample 2	14251618	14238188	33.46	14298244	14246769	33.5	14341847	14262678	33.43	14356382	14250132	33.42
	Sample 3	14275154	14217242	33.42	14240646	14111615	33.51	14320530	14166519	33.38	14312009	14138998	33.4
	Sample 4	14280549	14226612	33.4	14263763	14211021	33.34	14345200	14272900	33.2	14318448	14241382	33.17
	Sample 5	14268190	14182812	33.4	14218801	14102192	33.48	14287333	14155066	33.33	14304631	14158562	33.29
	Sample 6	14267575	14183870	33.41	14206012	14097160	33.5	14265963	14144176	33.37	14275276	14126308	33.33
	median	14268460.5	14221927	33.415	14252204.5	14161318	33.495	14319132.5	14208717.5	33.375	14315228.5	14199972	33.365
*C. gattii*	Sample 1	18374056	13963456	47.95	15618076	3018791	45.87	15848723	1127999	45.39	15649875	979468	45.62
	Sample 2	18373936	16738202	47.87	17275771	2829797	47.74	17401496	3193078	47.65	17335832	2478663	47.64
	Sample 3	18373817	16748750	47.87	17249122	2811973	47.77	17403823	2993154	47.66	17331969	2314973	47.69
	Sample 4	18373586	16911300	47.86	17292994	3406667	47.78	17435803	3916947	47.7	17395149	2892625	47.71
	Sample 5	18371784	17309929	47.88	17667739	10488842	47.95	17746664	10916558	47.91	17719423	10110195	47.82
	Sample 6	18374011	15590434	47.88	17093485	3649510	47.47	17283501	3129355	47.09	17341085	2347707	47.07
	median	18373876.5	16743476	47.875	17262446.5	3212729	47.755	17402659.5	3161216.5	47.655	17338458.5	2413185	47.665
*S. cerevisiae*	Sample 1	11900917	11786751	38.26	11756094	11627598	38.37	11762061	11614289	38.27	11770518	11610040	38.24
	Sample 2	11927452	11786979	38.22	11817583	11391169	38.31	11835515	11392663	38.24	11841970	11389993	38.23
	Sample 3	11867150	11717686	38.28	11714984	11611725	38.37	11728646	11591392	38.3	11734569	11542743	38.2
	Sample 4	12048365	11746218	38.27	11701491	11557641	38.31	11744219	11530244	38.26	11726032	11472823	38.12
	Sample 5	11848014	11728342	38.26	11844556	11579727	38.37	11847283	11568021	38.25	11841609	11544386	38.21
	Sample 6	11898828	11680204	38.27	11650537	11518215	38.35	11683391	11519687	38.23	11676382	11483435	38.13
	median	11899872.5	11737280	38.265	11735539	11568684	38.36	11753140	11549132.5	38.255	11752543.5	11513089	38.205
*P. falciparum*	Sample 1	23184099	23030452	19.3	22783133	22726603	19.63	23110345	23037187	19.36	23277887	23197642	19.16
	Sample 2	23244418	23191818	19.33	22846745	22827099	19.64	23103471	23077430	19.44	23251109	23206304	19.29
	Sample 3	23278091	23119804	19.27	22794879	22740838	19.59	23071068	22992830	19.36	23170782	23115122	19.2
	Sample 4	23266743	23186289	19.33	22843636	22817074	19.64	23082452	23052262	19.44	23222395	23183221	19.29
	Sample 5	23167744	22187311	19.55	22597393	22360095	19.53	22902857	22526387	19.36	22879437	22148919	19.29
	Sample 6	23193836	20645915	19.63	21278952	20848467	19.64	22021131	21099604	19.32	21995137	20265232	19.27
	median	23219127	23075128	19.33	22789006	22733720.5	19.635	23076760	23015008.5	19.36	23196588.5	23149171.5	19.28

QUAST-based assembly statistics including for *C. albicans*, *C. gattii*, *S. cerevisiae*, and *P. falciparum* assembled genomes with minimap2 pre- and post-read correction with flye, medaka, and racon. Bonferroni’s multiple comparison one-way ANOVA statistical analysis was performed to determine significant differences (*p* < 0.05, *p* < 0.001) existing among the different groups.

## Data Availability

Data are contained within the article and Appendix A.

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
