# Peer review of "Three Rounds of Read Correction Significantly Improve Eukaryotic Protein Detection in ONT Reads"

_microorganisms, 2024, doi:10.3390/microorganisms12020247_

Round 1

Reviewer 1 Report

Comments and Suggestions for Authors

While the author endeavors to enhance protein detection in eukaryotic assembly through three distinct sequence correction methods applied to ONT reads, the chosen evaluation criteria are flawed. The oversight of not considering the diploid nature of eukaryotic genomes is apparent. The author relies solely on total length to assert the completeness of the corrected sequences, an approach that is entirely inaccurate. Although the evaluation of total aligned reads may seem plausible, the authors found it to be statistically insignificant (lines 133-134). Additionally, aligning SRA sequencing reads to the four eukaryotic reference genomes (Table S2) was inappropriate, as these reads were not sequenced from those strains. The main results presented in Table 1 appear inconclusive; the use of total length and GC for evaluating read correction tools raises questions. Furthermore, the manuscript contains numerous inappropriate descriptions, such as the statement that reads were assembled using minimap2, but it aligns reads to reference genomes. The claim in Line 44 that “errors accumulate as sequencing depth increases” is a misinterpretation. Ref10 employed different assemblers, and the errors accumulated from a specific assembler. Although QUAST can be used to evaluate eukaryotic or fungal genome assemblies, the relevant parameters were not employed in this study.

Author Response

We appreciate the reviewer's feedback on these concerns. We agree that there hasn't been extensive review of benchmarking read correction tools for fungi and parasites in the literature. Hence, the criteria we used for evaluation might be open to debate. The assessment of the read correction tools in our manuscript considered various parameters, not just the total length highlighted by the reviewer.

We want to emphasize that metrics like QUAST (total length, aligned length, GC%) and BUSCO (genome completeness) are commonly used to evaluate bioinformatic tools for read assembly and correction, as supported by multiple studies (DOI: 10.1038/s41587-021-01108-x, DOI: 10.12688/f1000research.21782.4, DOI: 10.1186/s12864-018-5381-7, DOI: 10.1093/bib/bbac146). Therefore, we believe these assessments are valid.

Minimap2, the sequence alignment tool we used, aligns sequences to a reference genome. We employed the reference genomes in Table S2 to create the corresponding FASTA files.

Regarding the statement in line 44, cited by R10, we maintain that while it might be controversial, it accurately describes the performance of the de novo assemblers used in their study.

Reviewer 2 Report

Comments and Suggestions for Authors

The manuscript titled ‘Three rounds of read correction significantly improves ­eukaryotic protein detection in ONT reads’ by Safar, assessed read correction tools’ impact on eukaryotic genome sequencing using Oxford Nanopore technology. Medaka excelled in genome completeness, while racon improved gene detection significantly. Both racon and medaka outperformed flye in protein annotation. Three rounds of read correction notably influenced gene detection and protein annotation, highlighting the importance of assembly quality. This is a novel aspect in a field of intense research. Still some issues need to be clarified, as listed below, before the manuscript can be accepted for publication in Microorganisms.  

1.      In line 123, the author states that minimap2 outperforms read correction in terms of total length across the four species. However, upon examining the results post three rounds of correction, Table 1 indicates that only C. gattii and S. cerevisiae align with this statement. I recommend the author to clarify or provide additional details regarding the experimental results on total length, either by adjusting the description or offering a more detailed explanation.

2.      In line 126, the author compares the median total length. I recommend the author to clearly indicate the corresponding numerical values in the table for better clarity and accuracy.

3.      Please enlarge the text within each figure to ensure readers can clearly understand the information provided, especially in Figure 2, 3, 4, and 5. The figures appear blurry, and the distinctions between a, b, c, d are challenging to discern. Improved clarity in the figures will enhance the overall readability of the manuscript.

4.      Clarify the specific bioinformatics algorithms employed by the correction tools mentioned. Provide a brief overview of the strengths and limitations of these algorithms.

5.      Provide more details on the criteria used for benchmarking the read correction tools, including any specific datasets or metrics considered.

Author Response

Comments 1: In line 123, the author states that minimap2 outperforms read correction in terms of total length across the four species. However, upon examining the results post three rounds of correction, Table 1 indicates that only C. gattii and S. cerevisiae align with this statement. I recommend the author to clarify or provide additional details regarding the experimental results on total length, either by adjusting the description or offering a more detailed explanation.

Response 1: Thank you for pointing this out. We agree with this comment regarding minimap2 performance in aligning C. albicans genomes. After double checking the calculations, we found that minimap2 outer performs in total length across all species but C. albicans, however, the overall performance of minimap2 was significant against all three rounds of read correcting tools. Therefore, we have modified the statement in lines 123-125.

Comments 2: In line 126, the author compares the median total length. I recommend the author indicate the corresponding numerical values in the table for better clarity and accuracy.

Response 2: We agree with the review. We have, accordingly, added the median values to Table 1.

Comments 3:  Please enlarge the text within each figure to ensure readers can clearly understand the information provided, especially in Figure 2, 3, 4, and 5. The figures appear blurry, and the distinctions between a, b, c, d are challenging to discern. Improved clarity in the figures will enhance the overall readability of the manuscript.

Response 3: The figures quality was improved in the manuscript. The text within figures is now clearly visible when zooming-in the manuscript.

Comment 4: Clarify the specific bioinformatics algorithms employed by the correction tools mentioned. Provide a brief overview of the strengths and limitations of these algorithms.

Response 4: The introduction was modified accordingly, lines 59-64.

Comment 5: Provide more details on the criteria used for benchmarking the read correction tools, including any specific datasets or metrics considered.

Response 5: The criteria used to benchmark the read correction tools were added to the results and discussion section lines 124-127. We used default metrics as mentioned in the materials and methods, except for QUAST, in which we used the -LG parameter as stated in line 99.

Reviewer 3 Report

Comments and Suggestions for Authors

Safar et. al have reviewed existing ONT sequencing datasets for four different organisms, and found that read correction *after assembly* improves the accuracy of protein detection, even when filtering the input to limit to higher-quality reads only.

It's good to see that you've included datasets from different research groups, and are using consistent flow cell technologies and sequencing kits, but the publication date of the datasets is variable (at least for C. albicans, which is the only one I've looked at), which may skew your results:

* C. albicans *
1 SRR19900988 - 2022-06-29
2 SRR19901033 - 2022-06-29
3 SRR7874309  - 2018-09-18
4 SRR21528967 - 2022-12-19
5 SRR17110952 - 2021-12-08
6 SRR17110953 - 2021-12-08

This is a particular problem for Oxford Nanopore sequences where the raw data have not been re-basecalled with the same basecaller, because there are substantial accuracy improvements in sequences over time. Bearing in mind that re-doing the study with recent datasets is far too much work for manuscript updates (and it would make more sense to use R10.4.1 data in that case), it would be good to at least consider dataset publication date (or better, sequencing date) as a covariate to test whether there is an age effect on accuracy.

However... I notice also that you have appled a quality filter (>Q10), which should deal with the majority of these time-based issues.

In any case, a metadata table for the datasets would be good as well (indicating publication date, sequencing date (if available), caller model used (if available), total bases, median base-level accuracy, number of reads, etc.). It's curious that Sample4 for C. albicans has the lowest BUSCO scores, given that it has the most recent publication date, and according to their methods they used the super-accurate basecalling model.

The publications associated with these datasets should go into your references, as your publication wouldn't have been possible without the help of those datasets. I notice that at least Rizzo et. al (associated with SRR21528967) is not present (https://doi.org/10.1371/journal.pgen.1010576).

Thank you for including information about the most recent ONT technology developments regarding flow cells and accuracy (R10.4.1, Q20+); it's unusual to see this in ONT papers that I review. Unfortunately, the introduction of your paper (lines 41-43) repeats old accuracy information from many years ago (2021). Given that publications are often delayed information anyway, and ONT accuracies are increasing substantially over time, you should introduce accuracy with information from a recent reference (e.g. https://rrwick.github.io/2023/12/18/ont-only-accuracy-update.html). Accuracy is no longer a limitation of ONT sequencing, and including misleading statements like this will give people the wrong impression of the technology. I am aware that this "errors are a problem" concept is the entire point of your publication, but think you should at least emphasise more strongly that these conclusions relate to the discontinued R9.4.1 / LSK109 flow cell and kit combination. I think the general idea of the publication (i.e. that error correction improves protein detection) is good, and will be applicable even for Illumina-level accuracies.

In light of the attention that the authors have taken to minimise dataset accuracy variation, and the benefit of this study, I'd be happy to accept this manuscript for publication once the datasets used are included in the references (even if no other changes were made to the manuscript).

Ngā mihi,

 - David Eccles

Author Response

Comments 1: It's good to see that you've included datasets from different research groups, and are using consistent flow cell technologies and sequencing kits, but the publication date of the datasets is variable (at least for C. albicans, which is the only one I've looked at), which may skew your results:

* C. albicans *

1 SRR19900988 - 2022-06-29

2 SRR19901033 - 2022-06-29

3 SRR7874309  - 2018-09-18

4 SRR21528967 - 2022-12-19

5 SRR17110952 - 2021-12-08

6 SRR17110953 - 2021-12-08

This is a particular problem for Oxford Nanopore sequences where the raw data have not been re-basecalled with the same basecaller, because there are substantial accuracy improvements in sequences over time. Bearing in mind that re-doing the study with recent datasets is far too much work for manuscript updates (and it would make more sense to use R10.4.1 data in that case), it would be good to at least consider dataset publication date (or better, sequencing date) as a covariate to test whether there is an age effect on accuracy.

However... I notice also that you have appled a quality filter (>Q10), which should deal with the majority of these time-based issues.

In any case, a metadata table for the datasets would be good as well (indicating publication date, sequencing date (if available), caller model used (if available), total bases, median base-level accuracy, number of reads, etc.). It's curious that Sample4 for C. albicans has the lowest BUSCO scores, given that it has the most recent publication date, and according to their methods they used the super-accurate basecalling model.

Response 1: Thank you for pointing this out. We agree that having variable publication dates of the datasets might skew the results. However, we carefully collected these datasets to be as closely related as possible. Although the datasets were from different research groups, as mentioned by the reviewer, all the datasets were generated using R9 flowcell, and we applied a quality filter of Q10 on all reads – which eliminated the age effect on accuracy.

Regarding the metadata table, we agree with the reviewer's comments and therefore, the supplementary table S1 was modified accordingly.  

Comment 2: The publications associated with these datasets should go into your references, as your publication wouldn't have been possible without the help of those datasets. I notice that at least Rizzo et. al (associated with SRR21528967) is not present (https://doi.org/10.1371/journal.pgen.1010576).

Response 2: The publications associated with the datasets were added to the references list [ref 48-55]

Comment 3:  Thank you for including information about the most recent ONT technology developments regarding flow cells and accuracy (R10.4.1, Q20+); it's unusual to see this in ONT papers that I review. Unfortunately, the introduction of your paper (lines 41-43) repeats old accuracy information from many years ago (2021). Given that publications are often delayed information anyway, and ONT accuracies are increasing substantially over time, you should introduce accuracy with information from a recent reference (e.g. https://rrwick.github.io/2023/12/18/ont-only-accuracy-update.html). Accuracy is no longer a limitation of ONT sequencing, and including misleading statements like this will give people the wrong impression of the technology. I am aware that this "errors are a problem" concept is the entire point of your publication but think you should at least emphasise more strongly that these conclusions relate to the discontinued R9.4.1 / LSK109 flow cell and kit combination. I think the general idea of the publication (i.e. that error correction improves protein detection) is good and will be applicable even for Illumina-level accuracies.

Response 3: We thank the reviewer for the very informative comment. We agree that ONT accuracies is rapidly improving and that the new flow cells and Q20+ kits will introduce much better sequencing data. However, the sequencing data retrieved in this study were all sequenced using the R9 flow cell. Therefore, we modified lines 41-43 to state that the lower accuracy is when using the R9 flow cell.

Reviewer 4 Report

Comments and Suggestions for Authors

- Page 2, line 70-75:

I can understand if S.cerevisiae genome is deployed for this purpose, as it is the most-widely studies eukaryotic genome. So it will serve well as main model organism both in genomics and metabolomics.

However, what is the reason for invoking the others? If you choose P.facliparum, why not P.malariae, and P.vivax for instance? Please kindly state your justification and references about those genomes!  State also that you pick S.cerevisiae because it is the model organism. 

- Page 3, paragraph 3:

How did you generate and/or retrieve the translated protein, as INTERPROSCAN only recognize protein FASTA file? 

- Page 3, line 134-141:

What is the significance of having high or low GC%? What is the threshold? 

- Line 156-157:

Why you did not observe the same phenomenon in P.falciparum?

- Figure 2-5:

Figure 2-5 are too small, and the resolution is poor. Please kindly resolve the issue accordingly! 

Comments on the Quality of English Language

The manuscript will require another round of minor English proofreading. 

Author Response

Comment 1: Page 2, line 70-75:

I can understand if S.cerevisiae genome is deployed for this purpose, as it is the most-widely studies eukaryotic genome. So it will serve well as main model organism both in genomics and metabolomics.

However, what is the reason for invoking the others? If you choose P.facliparum, why not P.malariae, and P.vivax for instance? Please kindly state your justification and references about those genomes!  State also that you pick S.cerevisiae because it is the model organism.

Response 1: Thank you for bringing attention to this matter. The selection of these organisms was primarily motivated by the availability of high-quality sequencing data in the SRA-NCBI database through ONT methods. This choice was further supported by their significance as model organisms, as exemplified by S. cerevisiae, and their significance as infection organisms. A statement was added to the introduction as requested, in lines 78-81.

Comment 2: Page 3, paragraph 3:

How did you generate and/or retrieve the translated protein, as INTERPROSCAN only recognize protein FASTA file? 

Response 2: Thank you for this question. According to interproscan documentation, the -f / -fasta subcommand will return results for either protein sequenced or nucleotides sequences. In the introduction of the interproscan documentation, it is stated that “Users who have novel nucleotide or protein sequences that they wish to functionally characterise can use the software package InterProScan to run the scanning algorithms from the InterPro database in an integrated way. Sequences are submitted in FASTA format. Matches are then calculated against all of the required member database’s signatures and the results are then output in a variety of formats.”

Comment 3: Page 3, line 134-141:

What is the significance of having high or low GC%? What is the threshold?

Response 3: Thank you for this question. Understanding the CG% of a genome aids in various genomic analyses, gene prediction, and evolutionary studies and provides insights into the genome's structure, function, and stability. There is no fixed threshold for the GC%; it differs from organism to organism. In particular, low-GC reads have fewer errors than high-GC reads. Therefore, GC% indicates how robust a read correction can be.

Comment 4: Line 156-157:

Why you did not observe the same phenomenon in P.falciparum?

Response 4: Thank you for this comment. The actual bioinformatic reason for flye to perform better than racon in fungi but not parasites in genome completeness and duplication rates BUSCO analysis is beyond the scope of this paper.

Comment 5: Figure 2-5 are too small, and the resolution is poor. Please kindly resolve the issue accordingly! 

Response 5: Figures 2-5 were modified as requested.

Comment 6: The manuscript will require another round of minor English proofreading. 

Response 6: The manuscript underwent thorough proofreading. Necessary modifications were implemented in its entirety.

Round 2

Reviewer 1 Report

Comments and Suggestions for Authors

Most eukaryotes have diploid cells, medaka_consensus is a consensus read correction tool that can not be used to provide dipolid information. Total length or aligned length used in Table 1 can not be used to evalute the correction tools, especially when the samples were not exactly matched to the reference.

Author Response

For research article

Three rounds of read correction significantly improve eukaryotic protein detection in ONT reads.

Response to Reviewer 1 (Round 2) Comments

Comment 1: Most eukaryotes have diploid cells, medaka_consensus is a consensus read correction tool that can not be used to provide dipolid information. Total length or aligned length used in Table 1 can not be used to evalute the correction tools, especially when the samples were not exactly matched to the reference.

Response 1: Thank you for this comment. We understand that medaka previously had special subcommands for haploid and diploid variant calling (medaka_haploid_variant and medaka_variant); however, the diploid subcommand (medaka_variant) has been deprecated and replaced in the newer version of medaka with more accurate and better performance package – please rerefer to medaka docs on GitHub. In addition, the usage of medaka consensus has been widely used in the literature for eukaryotes’ assemblies, which enhanced the accuracy of the consensus FASTA files:

1- Ohta, A., Nishi, K., Hirota, K. et al. Using nanopore sequencing to identify fungi from clinical samples with high phylogenetic resolution. Sci Rep 13, 9785 (2023). https://doi.org/10.1038/s41598-023-37016-0

2- Yu, PL., Fulton, J.C., Hudson, O.H. et al. Next-generation fungal identification using target enrichment and Nanopore sequencing. BMC Genomics 24, 581 (2023). https://doi.org/10.1186/s12864-023-09691-w

3- Sigova, E.A.; Pushkova, E.N.; Rozhmina, T.A.; Kudryavtseva, L.P.; Zhuchenko, A.A.; Novakovskiy, R.O.; Zhernova, D.A.; Povkhova, L.V.; Turba, A.A.; Borkhert, E.V.; et al. Assembling Quality Genomes of Flax Fungal Pathogens from Oxford Nanopore Technologies Data. J. Fungi 2023, 9, 301. https://doi.org/10.3390/jof9030301

4- Petersen C, Sørensen T, Westphal KR, Fechete LI, Sondergaard TE, Sørensen JL, Nielsen KL. High molecular weight DNA extraction methods lead to high quality filamentous ascomycete fungal genome assemblies using Oxford Nanopore sequencing. Microb Genom. 2022 Apr;8(4):000816. doi: 10.1099/mgen.0.000816